# Associations between the Bacterial Composition of Farm Bulk Milk and the Microbiota in the Resulting Swedish Long-Ripened Cheese

**DOI:** 10.3390/foods12203796

**Published:** 2023-10-16

**Authors:** Li Sun, Annika Höjer, Monika Johansson, Karin Hallin Saedén, Gun Bernes, Mårten Hetta, Anders H. Gustafsson, Johan Dicksved, Åse Lundh

**Affiliations:** 1Department of Molecular Sciences, Swedish University of Agricultural Sciences, SE-750 07 Uppsala, Sweden; monika.johansson@slu.se (M.J.); ase.lundh@slu.se (Å.L.); 2Norrmejerier Ek. Förening, Mejerivägen 2, SE-906 22 Umeå, Sweden; annika.hojer@norrmejerier.se (A.H.); karin.hallin-saeden@norrmejerier.se (K.H.S.); 3Department of Animal Nutrition and Management, Swedish University of Agricultural Sciences, SE-901 83 Umeå, Sweden; gun.bernes@slu.se (G.B.); marten.hetta@slu.se (M.H.); 4Växa Sverige, Ulls väg 29 A, SE-751 05 Uppsala, Sweden; anders.h.gustafsson@vxa.se; 5Department of Animal Nutrition and Management, Swedish University of Agricultural Sciences, SE-750 07 Uppsala, Sweden; johan.dicksved@slu.se

**Keywords:** farm and dairy silo milk microbiota, starter and non-starter lactic acid bacteria, *Lactobacillus*, *Lactococcus*, cheese ripening

## Abstract

The maturation of a traditional Swedish long-ripened cheese has shown increasing variation in recent years and the ripening time is now generally longer than in the past. While the cheese is reliant on non-starter lactic acid bacteria for the development of its characteristic flavour, we hypothesised that the observed changes could be due to variations in the microbiota composition and number of bacteria in the raw milk used for production of the cheese. To evaluate associations between microbiota in the raw milk and the resulting cheese, three clusters of commercial farms were created to increase variation in the microbiota of dairy silo milk used for cheese production. Cheese production was performed in three periods over one year. Within each period, milk from the three farm clusters was collected separately and transported to the cheese production facility. Following pasteurisation, the milk was processed into the granular-eyed cheese and matured at a dedicated cheese-ripening facility. For each cheese batch, farm bulk and dairy silo milk samples, a starter culture, early process samples and cheese samples from different stages of maturation (7–20 months) were collected and their microbiota characterised using 16S rRNA amplicon sequencing. The microbiota in the farm bulk milk differed significantly between periods and clusters. Differences in microbiota in dairy silo milk were observed between periods, but not between farm clusters, while the cheese microbiota differed between periods and clusters. The top 13 amplicon sequence variants were dominant in early process samples and the resulting cheese, making up at least 93.3% of the relative abundance (RA). *Lactococcus* was the dominant genus in the early process samples and, together with *Leuconostoc*, also dominated in the cheese samples. Contradicting expectations, the RA of the aroma-producing genus *Lactobacillus* was low in cheese during ripening and there was an unexpected dominance of starter lactic acid bacteria even at the later stages of cheese ripening. To identify factors behind the recent variations in ripening time of this cheese, future studies should address the effects of process variables and the dairy environment.

## 1. Introduction

The production of long-ripened extra-hard cheese is a complex process in which microbial and biochemical processes contribute to the characteristic flavour and texture of the cheese [1]. The metabolic activity in the cheese core during ripening derives from native enzymes in the milk, added rennet, added lactic acid bacteria (LAB) starter culture and adventitious non-starter lactic acid bacteria (NSLAB). The major LAB metabolic pathways involved in flavour formation in cheese during ripening can be categorised into primary (lipolysis, proteolysis and metabolism of residual lactose, lactate and citrate) and secondary (metabolism of fatty acids and amino acids) pathways [1]. Proteolysis is of the greatest importance for the final texture and flavour of the cheese, and peptides and amino acids resulting from activities of the starter culture provide the main nutritional compounds for NSLAB. The amino acids also act as precursors in a series of catabolic reactions, resulting in volatile aroma compounds, e.g., alcohols, aldehydes, ketones, esters and phenolic and sulphur compounds [2]. 

The NSLAB in cheese is believed to originate from the milk, either by surviving pasteurisation or by contaminating the pasteurised milk later in the manufacturing process via the dairy environment, e.g., a facility-specific “house” microbiota [3]. NSLAB is a heterogeneous group of mesophilic bacteria consisting of, e.g., *Lactobacillus*, *Pediococcus* and *Enterococcus*. Species of facultative heterofermentative lactobacilli, e.g., *Lactobacillus casei*, *Lactobacillus paracasei*, *Lactobacillus plantarum*, *Lactobacillus curvatus* and *Lactobacillus rhamnosus*, predominate in Cheddar-type cheeses [4]. *L. rhamnosus*, *L. casei* and *L. paracasei*, commonly referred to as the *L. casei* group, are highly abundant during the ripening of extra-hard, cooked cheeses such as Grana Padano [5]. These bacteria are present in low numbers and do not grow well in raw milk, but dominate the viable bacterial population in the maturing cheese [6]. However, dependence on adventitious NSLAB introduces variability into the ripening process, resulting in differences between cheeses produced at the same cheese-making plant on different days, and even in different batches on the same day [7].

In recent years, there has been increasing variation in the maturation of a Swedish traditional long-ripened cheese and the ripening time is now generally longer than in the past. Stricter hygiene on farms, including pre-milking, udder-cleaning, and teat-disinfection routines, have likely contributed to reducing the total number of bacteria in bulk milk and therefore also the number of NSLAB in the milk, with consequences for the cheese maturation process. In a recent study, we explored farm-related factors contributing to the variation in the microbial community in bulk milk from different dairy farms [8]. The results revealed an effect of routines associated with teat preparation and cleaning of the milking equipment on bulk milk microbiota, with milk from farms using an automated milking system (AMS) having different microbial composition than milk from tie-stall farms. We also observed a difference in the microbial composition between milk from AMS farms using different brands of milking robots, which was likely explained by differences in the performance of the robots. In contrast, Doyle et al. [9] concluded that teat preparation has a limited impact on raw milk microbiota and that the herd habitat is the major driver of milk microbiota composition. Gagnon et al. [10] investigated silage as a contamination source of facultative heterofermentative LAB in milk and concluded that silage is probably a minor contributor. We found similar results in a recent study (unpublished data). In practice, NSLAB are difficult to control and their pathways to the milk are diverse. For this reason, knowledge of the diversity and abundance of NSLAB species and factors affecting their presence in the raw milk is essential for the successful production of long-ripened cheese.

The aim of this study was to analyse the microbiota in bulk milk and determine its contribution to the microbiota in dairy silo milk and in the resulting long-ripened cheese.

## 2. Materials and Methods

### 2.1. Participating Farms

The participating farms were located in a region between 64°1′ to 64°9′ N and 20°5′ to 21°4′ E in the county of Västerbotten, Sweden. All selected farms routinely delivered their milk to the participating cheese-making facility. To introduce greater variation in the milk used for cheese-making, three different clusters of farms (A, B and C) producing milk differing in various aspects were created. In total, 18 farms were selected based on data on detailed milk composition and farm characteristics obtained in a previous study [11]. Cluster A consisted of farms delivering milk of average quality in terms of composition, total bacteria count and microbiota. Cluster B consisted of mostly tie-stall farms delivering milk characterised by higher fat and protein content and fewer clostridia. Milk from farms in cluster C typically had higher total numbers of bacteria and larger proportions of lactic acid bacteria (Table 1).

### 2.2. Cheese Production Practice

Cheese production was performed in three periods: November 2017 (period 1), February–March 2018 (period 2) and September 2018 (period 3). In each period, bulk milk from the farms belonging to each of the three clusters was collected separately and transported to a dedicated dairy silo at the cheese-making facility. This was repeated on two separate days during one week per period. The total volume of dairy silo milk originating from each farm cluster was approximately 15,000 L, which was the volume needed for full-scale production of one batch of cheese. After pasteurisation (72 °C, 15 s), the milk was processed into a Swedish traditional, granular-eyed long-ripened cheese, using a mesophilic starter and a coagulant consisting of 75/25 bovine chymosin and pepsin (180 IMCU, Sacco System Nordic [Kemikalia], Skurup, Sweden). Cheese making comprised long cooking periods (several hours) at temperatures above 40 °C. The cheeses were produced in 18 kg cylinders (16 cm high), brine-salted to a salt content of around 1.2%, waxed and ripened at specific temperatures between 10° and 13 °C at a dedicated cheese-ripening facility. Thus, in each of the three periods, milk from each farm cluster was used for one batch of full-scale cheese production on two separate days, resulting in a total of 18 cheese batches (3 periods × 3 clusters × 2 production days × 1 batch per day). 

### 2.3. Sampling of Farm Bulk Milk, Dairy Silo Milk and the Resulting Cheese

On each day of cheese production, individual farm bulk milk (250 mL) was sampled by the tanker driver during milk collection on the farms. These milk samples were kept in a cool box during transportation to the dairy facility. The raw dairy silo milk (250 mL) used for cheese production was sampled before processing. During cheese production, samples of freshly propagated bulk starter (50 mL), milk gel (50 mL) and cheese grains before pressing (40–50 g) were collected in the early cheese-making process (early process samples, EPSs). Fresh cheese at 24 h, cheese matured for 7 months and cheese matured for 12–20 months (with 2-month intervals) were also sampled. On each day of production, approximately 80 cheese wheels were produced from the milk collected from a farm cluster and of these, at least four cheese wheels produced in the middle of the batch were reserved for this study. On each cheese sampling occasion, one drill core of cheese (25–30 g) was extracted for microbiota analysis. An individual cheese was never sampled on more than four occasions. 

The farm bulk and dairy silo milk samples were stored at 4 °C at the dairy facility until transportation to SLU, Uppsala, Sweden. Samples were transported at ambient temperature using cooling pads, and upon arrival, all milk samples were aliquoted into 2-mL tubes and stored at −80 °C until DNA extraction. The maximum time from sampling of bulk and silo milk to storage at −80 °C was 24 h. The EPSs and cheese samples were stored at −60 °C at the dairy facility until transportation to SLU, Uppsala. Upon arrival, all samples were stored at −80 °C until DNA extraction. In addition, fresh cheese samples aged for 12 to 20 months (with 2 months interval) were transported refrigerated to SLU, Uppsala, and stored at 4 °C until analysis of total bacteria count. The maximum time from sampling of drill cores of the cheese to analysis of the fresh samples was 48 h. Due to practical circumstances, the EPSs and fresh cheese samples at 24 h were missing in period 2 for cluster B on cheese production day 2.

### 2.4. Total Bacteria Count in Farm Bulk Milk, Dairy Silo Milk and Cheese

Total bacteria count in farm bulk milk was analysed at Eurofins Food & Feed Testing Sweden AB (Jönköping, Sweden) using Bactoscan FC (Foss, Hilleroed, Denmark), while bacteria in the dairy silo milk samples were determined by culturing on plate count agar (PCA; Casein-peptone Dextrose Yeast Agar, Merck KGaA, Darmstadt, Germany) followed by incubation at 30 °C for 3 days at the cheese-making facility. Total bacteria count in fresh cheese samples was analysed at SLU, Uppsala. In brief, 25 g of cheese was homogenised in 100 mL of phosphate-buffered saline (PBS, pH 7.4) for 2 min at normal speed in a stomacher bag (400 Classic, Seward, West Sussex, UK) in a stomacher blender (Stomacher 400, Seward, AK, USA) at room temperature. Decimal dilutions in PBS were prepared, 0.1 mL aliquots of the appropriate dilutions were inoculated in duplicate on plate count agar and incubated at 30 °C for 48 h to obtain viable counts as colony forming unit (cfu) per gram of cheese.

### 2.5. DNA Extraction from Bacteria in Milk and Cheese

Liquid samples (milk, starter culture and milk gel) were thawed at 37 °C for 15 min in a water bath. Solid samples (cheese granules, cheese of different ages) were thawed at room temperature for 15 min. Milk gel and solid cheese samples (25 g) were homogenised as previously described for analysis of total bacteria count. For cheese grains before pressing and cheese aged for 24 h, the homogenisation time was extended to 6 min. DNA extraction was conducted using a PowerFood DNA isolation kit (Qiagen AB, Sollentuna, Sweden) according to a customised protocol. In brief, a 1.8 mL portion of whole milk or a homogenised aliquot was centrifuged at 13,000× *g* for 15 min at 4 °C, and then incubated on ice for 5 min. The resulting cell pellets with carefully collected fat layer were resuspended in 450 μL of MBL buffer (provided with kit). The resuspended mixture was transferred to MicroBead tubes (provided with kit). Cell lysis was conducted by incubating the tubes at 65 °C for 10 min, after which they were processed in a Fastprep 24 instrument (MP Biomedicals, Santa Ana, CA, USA) at 5.0 speed for 60 s, repeated two times with a 5-min pause. The tubes were then centrifuged at 13,000× *g* for 15 min at 4 °C, followed by incubation on ice for 5 min. The supernatant excluding the fat layer was transferred to new 2 mL collection tubes and the remaining steps were carried out according to the manufacturer’s protocol. The resulting DNA was eluted with 50 μL of buffer EB (provided with kit) and stored at −20 °C until use.

### 2.6. Illumina Amplicon Library Construction, Sequencing and Bioinformatic Analysis

The DNA extracted from the milk, EPSs and cheese samples was used to construct a 16S rRNA gene library with primers 515F and 805R [12]. Illumina adaptors and barcodes were used for amplification, following a two-step PCR approach previously described [13]. The 16S rRNA gene library was sequenced using the Illumina Miseq platform at SciLifeLab (Stockholm, Sweden). The raw sequencing data have been deposited in the Sequence Read Archive at the National Center for Biotechnology Information database (http://www.ncbi.nlm.nih.gov/sra, (accessed on 10 September 2023)), under accession number PRJNA1010645. Bioinformatic data processing was performed using Quantitative Insights into Microbial Ecology 2 (Core 2020.11) [14]. The raw demultiplexed reads were trimmed using Cutadapt to remove primer sequences [15]. Sequencing base with quality score below 30 was trimmed off from the 3′ end. A read was discarded if it contained N base or did not contain primer sequences. The trimmed reads were further processed using DADA2 to de-noise, de-replicate reads, merge pair end reads and remove chimeras [16], using a truncation length of 210 and 160 bp for forward and reverse reads, respectively. A phylogenetic tree was built using FastTree and MAFFT alignment [17,18]. The SSU Ref NR 99 138 dataset was first trimmed to the corresponding primer region and used for training the classify-sklearn taxonomy classifier [19,20,21]. Amplicon sequence variants (ASVs) were assigned taxonomy using the resulting classifier. The weighted UniFrac distance matrix and alpha rarefaction were generated using the QIIME2 diversity plugin [14]. To identify the dominant ASVs present in farm milk, silo milk, EPS and cheese, the top 30 ASVs with relative abundance (RA) higher than 0.04% in the total sequencing pool were selected for analysis.

### 2.7. Statistical Analysis

Microbiota analyses were performed with the q2-diversity plugin. The rarefied ASV table was used to calculate the number of observed ASVs. Kruskal–Wallis rank test [22] and Benjamini and Hochberg (B-H) correction [23] were used to identify statistical differences in RA of the ASVs between groups, i.e., periods and clusters. Principal coordinate analysis (PCoA) was used to visualise differences in microbial composition based on the generalised UniFrac distances. Permutational multivariate analysis of variance (PERMANOVA) testing of generalised UniFrac distance matrix with (B-H) correction [24] was conducted to evaluate differences between groups. 

## 3. Results

The alpha diversity of the microbiota in farm bulk and dairy silo milk was higher (*p* < 0.001) than that of the microbiota in EPSs and cheese in terms of the number of observed ASVs (Figure 1a) and the Shannon index (*p* < 0.001). The beta diversity of the microbiota present in farm bulk and dairy silo milk was also different from that in EPSs and cheese (*p* < 0.001), as revealed using the generalised UniFrac distance (Figure 1b).

### 3.1. Exploring the Microbiota Present in Farm Bulk and Dairy Silo Milk

The microbiota in farm bulk milk showed significant differences between periods and clusters (Figure 2a,b, both *p* < 0.001). A pairwise comparison confirmed that each period was different from the others (*p* < 0.001). In the pairwise comparison of the microbiota in bulk milk from the different clusters, a difference was observed between clusters A and B (*p* < 0.05), and between clusters B and C (*p* < 0.01). However, there was no difference between clusters A and C (*p* > 0.05) and no clear separation in the PCoA plot (Figure 2b). On comparing the microbiota in the dairy silo milk samples, differences between periods were still evident (Figure 3, *p* < 0.05). However, in the pairwise comparison of the microbiota in dairy silo milk during different periods, the only significant difference observed was between periods 1 and 2 (*p* < 0.05). There was no significant difference in microbiota in dairy silo milk between the farm clusters.

The microbiota in the dairy silo milk resulted from a combination of volume, total bacteria count and microbiota composition of the bulk milk delivered from the individual farms in the same cluster. This was illustrated in Figure 4 for farm no. 20 (F20) in cluster C, with an elevated total bacteria count and a high relative abundance (RA) of *Streptococcus* bd2e in the bulk milk used on the second cheese-making day (D2) in period 1. Since the volume of milk delivered from this farm made up 40% of the total volume of the dairy silo, *Streptococcus* bd2e also had a high RA in the dairy silo milk (SM). 

### 3.2. Exploring the Microbiota in Cheese

The PCoA of generalised UniFrac distance showed significant differences in cheese microbiota between the periods (Figure 5a, *p* < 0.001), and the pairwise comparison confirmed that all periods differed from each other (*p* < 0.001). There was also a significant difference in cheese microbiota between the farm clusters (Figure 5b, *p* < 0.001), with pairwise comparisons confirming that all clusters differed from each other (*p* < 0.01).

The total bacteria count in cheese samples aged for 12 to 20 months varied from log 2.82 to 6.74 cfu/g cheese (Figure 6). There was fluctuation in total bacteria counts in cheese during this stage of maturation, but there was a clear trend suggesting that cluster A had a lower total bacteria count in all three periods.

To investigate the development of the microbiota in the early stages of the cheese-making process and during ripening, samples of the starter culture, milk gel, cheese grains and cheese aged for 24 h and 7–20 months were analysed (Figure 7). The top 13 ASVs were dominant in both early process and cheese samples, making up at least 93.3% of the RA. *Lactococcus* was the dominant genus in EPSs and in cheese aged for 24 h. In cheese samples aged for 7 to 20 months, *Lactococcus* and *Leuconostoc* were the two dominant genera. In some batches of cheese, *Leuconostoc* already started to increase in RA in the young cheese (24 h). In contrast, the genus *Lactobacillus* was not among the most abundant genera in most samples. However, this genus was detected in all cheeses, on at least one sampling occasion for each batch of cheese and at a higher RA in some batches of cheese, especially in period 1. *Streptococcus* was more common in cheese from cluster C and the RA of this genus was higher in cheese from periods 1 and 3. *Acinetobacter* was occasionally detected, with a higher RA in one batch of cheese produced in period 1 (cluster C). Several ASVs within the genera *Lactococcus*, *Leuconostoc* and *Lactobacillus* were found. Within the genus *Lactobacillus*, ASV c982 was the most commonly observed ASV, while ASV 2653 and 3486 were present at a high RA in a few cheese batches, in most cases associated with cluster B. *Lactococcus* b39c was dominant compared with the other four ASVs in the same genus, i.e., d90e, 072f, bb15 and cc1f. It is worth noting that *Lactococcus* bb15 and cc1f were more commonly found in periods 1 and 2. For the genus *Leuconostoc*, in most batches ASV 7f62 had a higher RA than ASV 24cd, although the latter was present at a high RA in some of the batches, e.g., in period 2, cluster B.

### 3.3. Common ASVs Found in Different Sample Types

Nine of the top 13 ASVs found in early process and cheese samples (Figure 7) were also identified in the dairy silo milk samples (Figure 8). Most of the ASVs that were common in cheese had rather low RA in the milk. *Streptococcus* bd2e was the exception, and this ASV had a higher RA in dairy silo milk samples from cluster C than in samples from the other two clusters in all three periods. As previously mentioned, cheese samples associated with cluster C had a higher RA of this ASV. The four ASVs observed in cheese but not in dairy silo milk were *Acinetobacter* 6933, *Lactobacillus* 2653, *Lactococcus* bb15 and *Lactococcus* 072f.

To avoid the risk of overlooking ASVs common to both silo milk and cheese due to sequencing limitations, e.g., poor sequencing depth, screening for the ASVs common to cheese and the associated farm bulk milk, dairy silo milk and EPSs was conducted (Table 2). In this screening, the *Lactobacillus* 0b36, *Lactobacillus* 57c1 and *Tetragenococcus* 49a8 present in the cheese were also found in the farm bulk milk, but not in the dairy silo milk or EPS, and only at a very low frequency. Screening for the ASVs present in both EPSs and cheese showed that the previously mentioned *Lactococcus* bb15 and *Lactococcus* 072f were present at high frequencies in EPSs, including in the starter culture. *Acinetobacter* 6933 and *Lactobacillus* bf19 were identified in EPSs, but not in farm bulk or dairy silo milk, and were limited to one batch of production (period 1, cluster C) where *Lactobacillus* was identified in the starter culture. However, *Lactobacillus* 2653, *Lactococcus* 6fa5 and *Lactococcus* ffd9 were only identified in the cheese.

## 4. Discussion

In this study, 16S rRNA gene amplicon sequencing was used to investigate the microbiota of raw bulk milk collected at farms, the resulting silo milk at the cheese-making facility, samples collected early in the cheese-making process and samples taken repeatedly during cheese ripening in order to investigate changes in the composition of the microbiota throughout the cheese manufacturing process. A rarefaction curve and Shannon index values revealed that the microbiota in farm bulk and dairy silo milk samples had a much higher alpha diversity than that in EPSs and cheese samples (Figure 1). The low alpha diversity in EPSs and cheese samples in our study is a consequence of the dairy processes, i.e., pasteurisation of the raw milk and the fermentation taking place after addition of the starter culture. Species that cannot compete with the active lactic acid bacteria, which ferment lactose into lactic acid, thereby lowering pH, will be eliminated and disappear. This was also illustrated by Choi et al. [25], who investigated the overall microbial community shift during Cheddar cheese production from the raw milk to the aged cheese. SLAB inoculation decreased the microbial richness by inhibiting the growth of other bacteria present in the milk. In contrast, species richness increased with ripening time for the non-pasteurised Parmigiano Reggiano [26]. 

Comparing the diversity associated with farm bulk and dairy silo milk, the higher number of ASVs in dairy silo milk was probably explained by the fact that the dairy silo comprised bulk milk from several farms, with each farm contributing a different microbiota. A higher alpha diversity of the microbiota in farm bulk milk than in samples representing the subsequent stages in cheese production was previously reported by McHugh, et al. [27], who explored changes in the dairy microbiome from the farm through transportation and processing to skimmed milk powder. Although the microbiota composition of the bulk milk was not the only criterion used to create the three farm clusters in the present study, the microbiota in the bulk milk from cluster B was different from that in the bulk milk from clusters A and C (Figure 2). The period had a more pronounced effect on the bulk milk microbiota, with each period being different from the other two in this respect. Similarly, Skeie et al. [28] reported shifts in the microbiota composition of raw farm bulk milk during a three-month monitoring period and suggested random variation due to variable sources of contamination at the farm level. A follow-up study on the same farms two years later concluded that temporal within-farm changes in bulk milk microbiota are mostly driven by mastitis-related genera [29]. For some farms, there were major shifts over time in milk microbiota which were not correlated with changes in management, indicating that other factors, e.g., the weather during the harvesting season, contributed to the observed differences [29]. Investigating the variation in the microbiota in bulk milk collected monthly on 42 dairy farms over one year, we found that the type of dairy farm (milking system) had a strong effect on the bulk milk microbiota, an effect likely associated with hygiene routines during the pre-milking and cleaning of the milking equipment [8]. The 18 farms used in the present study were selected from among those original 42 farms and an effect of milking system on bulk milk microbiota was also clear in the present study.

In contrast to farm bulk milk, there were no significant differences in the microbiota in dairy silo milk from the three farm clusters, while the difference between periods persisted at the dairy silo level (Figure 3). Under certain conditions, the microbiota composition of the dairy silo milk was strongly affected by the farm bulk milk microbiota. This is likely explained by differences in the volumes of milk delivered to the dairy by the individual farms. It was particularly obvious in period 1 (Figure 4), where one farm belonging to cluster C contributed a large proportion of the milk in the dairy silo, and this had a clear impact on the dairy silo milk microbiota. While *Streptococcus salivarius* ASV bd2e was dominant in the bulk milk from this farm, it also became dominant in the dairy silo milk, and it was one major ASV in the resulting cheese. This highlights the importance of good herd management, especially on larger farms, to maintain high hygiene standards and keep the total bacteria count in the bulk milk at low levels. The genera *Romboutsia*, commonly associated with the intestinal tract, and *Staphylococcus*, a major mastitis pathogen associated with the mammary gland tissue, had a relatively high RA in the bulk and silo milk samples, but had low RA (<0.1%) in the cheese samples, which was explained by the pasteurisation of the raw milk before cheese making.

Although there was no clear effect of farm cluster on the microbiota of the dairy silo milk used for cheese production, the cheese microbiota showed significant differences among both periods and clusters (Figure 4). In general, *Lactobacillus*, a NSLAB species known from previous studies to be important for flavour development in this traditional Swedish cheese, was present at a higher RA in cheeses produced in period 1 compared with periods 2 and 3 (Figure 7). *Acinetobacter* was also more frequently observed in cheese produced in period 1 associated with cluster C and day 1. Considering that the *Acinetobacter* ASV 6933 was not detected in our farm or silo milk samples, contamination may have occurred in the dairy environment. Within the genus *Lactococcus*, a single ASV was completely dominant in period 3, while several minor ASVs of this genus were observed in periods 1 and 2. Screening of farm bulk milk, dairy silo milk and EPSs to identify ASVs common to both milk and cheese revealed that the dominant ASVs found in cheese (average RA > 1%) were also present in milk and EPS samples (Table 2, Figure 7 and Figure 8). This confirms that the LAB of importance for the cheese manufacturing process are common in both farm and dairy environments. However, the dominant LAB species found in cheese were usually present at low RAs in farm bulk and dairy silo milk, while NSLAB, specifically *Lactobacillus*, showed very low RA in both milk and EPSs. It is, therefore, difficult to draw conclusions regarding the origin of the *Lactobacillus* ASV found in the cheese; it could stem from the raw milk, the dairy environment or both. More sensitive methods, e.g., qPCR using primers targeting specific NSLAB, are needed for the identification of the origin of these bacteria. 

The starter LAB were present in the EPS and cheese samples at a relatively high RA. Some variation in the starter LAB was observed, e.g., the three less dominant *Lactococcus* ASVs (d90e, 072f and bb15) were present at much lower RAs in the cheese from period 3 than in the cheese from periods 1 and 2 (Figure 7). Different steps in the cheese-making process may have a major effect on the cheese microbiota, including changes in the abundance of NSLAB. As an example, a previous study by Porcellato and Skeie [30] reported that a small change (2 °C) in scalding temperature can significantly influence the cheese microbiota. Stress and bacteriophage attack may also give rise to variation in the LAB starter between different production periods, illustrating the importance of using starter cultures with high strain diversity, e.g., undefined, mixed-strain starter cultures, which are generally considered more resilient [31]. *Lactococcus* d90e was identified in the screening of ASVs passing from milk to cheese and from EPSs to cheese, while 072f and bb15 were only observed in EPSs and cheese. The identification of ASVs which were only associated with EPSs suggests a potential enrichment of specific LAB in the cheese production process, e.g., contamination by facility-specific “house” microflora from processing equipment and the dairy environment [32]. According to Bokulich and Mills [3], this “house” microbiota plays an important role in shaping site-specific product characteristics. In addition, three other ASVs (*Lactobacillus* 2653, *Lactococcus* 6fa5 and *Lactococcus* ffd9) were only detected in cheese, and were present neither in milk nor in EPS samples. 

Comparing cheese aged for 12 months or longer, cheese from cluster A milk had a consistently lower total bacterial count in all three periods (Figure 5). The numbers of viable bacteria in the cheese were generally lower than expected, in most samples ranging between 10^5^ and 10^6^ cfu/g cheese and in cluster A samples exhibited even lower counts, i.e., 10^3^–10^5^ cfu/g. In a previous study on the same type of cheese [33], the number of LAB cultured on MRS agar was in the order of 10^6^–10^7^ cfu/g cheese from 12 weeks until 56 weeks of ripening. The use of plate count agar instead of MRS for bacterial counting in our study likely explains the difference in observed CFU levels. The sequencing results revealed that cheese from cluster A had a lower RA of *Lactobacillus* in all periods, but especially in periods 2 and 3 (Figure 7). This suggests that the low number of viable bacteria in cheese from cluster A was likely associated with a low number of NSLAB. 

A decline in starter LAB and proliferation of NSLAB usually take place during the early stage of ripening. In a culture-based study on Cheddar cheese, Dasen et al. [34] found that the *Lactobacillus* count on *Lactobacillus*-selective medium reached a peak after only 3 months, while counts on M17 medium declined from the beginning of ripening, reaching the lowest value at around 9 months. In a study by Rehn et al. [33], starter LAB counts were higher than NSLAB counts up to 3 weeks of ripening and decreased between 3 and 12 weeks, while after 26 weeks of ripening NSLAB dominated in the cheese. The high RA of starter LAB (i.e., *Lactococcus* and *Leuconostoc*) in cheese from 7 until 20 months of ripening was therefore unexpected (Figure 7). Although several studies have confirmed the presence of viable *Lactococcus* at later stages of cheese ripening using RNA-based analysis [35,36], their contribution at this stage needs further investigation. By using retentostat cultures, van Mastrigt and co-workers mimicked the cheese ripening conditions and quantified the aroma profile of *Lactococcus lactis* [37,38]. One possible explanation for the high abundance of starter LAB in our study could be that the DNA from dead starter bacteria was amplified and sequenced. In a study examining the effects of the cheese-making process on the cheese microbiota in a Dutch-type cheese, treating bacteria cells with PMA (propidium monoazide) to selectively detect viable bacteria, Porcellato and Skeie [30] found that the RA of *Lactobacillus* spp. increased and the RA of *Lactococcus lactis* decreased in cheese ripened for 1 and 3 months. Barzideh et al. [39] found that after PMA treatment the RA of *Lactococcus* in a long-ripened Cheddar cheese was below 20% after 7 months, whereas non-PMA treated *Lactococcus* had RA of 30% at the same timepoint. In our study, *Leuconostoc* reached high RA at month 7 of ripening and in two batches had already increased to 25% and 39% RA, respectively, at 24 h (Figure 7, period 3, clusters A and B). Although *Leuconostoc* was included in the DL-type starter culture used in this study, its growth dynamic during cheese ripening was similar to that of a typical NSLAB, in agreement with observations in the case of Cheddar cheese [39]. 

The RA of *Streptococcus* (ASV bd2e) increased during ripening, and it was present at a higher RA in the cheese from cluster C (Figure 7). It was also present at a high RA in the bulk milk from one of the larger individual farms in cluster C; milk from this farm contributed with 40% of the total volume of dairy silo milk in period 1 (Figure 4). Some strains of *Streptococcus*, e.g., *S*. *thermophilus,* are used in thermophilic starter cultures for the production of different types of cheese [40,41]. Since a mesophilic starter culture was used to produce the cheese, the *Streptococcus* most likely originated from raw bulk milk, surviving pasteurisation, and suggesting the passage of the *Streptococcus* ASV from farm bulk to dairy silo milk, EPSs, and the resulting cheese (Table 2). According to Johnson et al. [42], the *Streptococcus* group is commonly found in aged Cheddar cheese owing to the long production schedule in modern cheese manufacturing, which creates conditions that support the growth of microorganisms in the processing environment. This may introduce differences between cheese batches produced early and late during the same day, due to bacterial growth on food contact surfaces and changes in the starter culture [42]. Due to the short read length of the sequencing platform used, the type of *Streptococcus* could not be identified in detail, so we cannot be sure that passage of the ASV truly reflected the transfer of this group of bacteria from bulk milk to cheese. It is also worth noting that due to the short read length, observing the same ASV in different sample types did not necessarily confirm the presence of the same bacteria strain. 

In contrast to our hypothesis that the increasing variation and longer ripening time of the investigated cheese would reflect the variation in the raw milk microbiota, the results showed little association between the milk microbiota and cheese ripening. Future studies should therefore address facility-specific aspects, e.g., by characterising the in-house microbiota, isolating and characterising the NSLAB from successful cheeses, evaluating the possibility to use isolated NSLAB as an adjunct culture or assessing the effect of slight differences in heat treatment of the milk used for cheese making.

## 5. Conclusions

Through careful selection of participating farms, we managed to create three clusters with some variation in bulk milk microbiota between clusters, but this variation was less apparent in the resulting dairy silo milk. Variation over time, i.e., between periods, was more profound, both in farm bulk and dairy silo milk. The cheese microbiota varied between periods and clusters, but the variation showed little association with the microbiota in the dairy silo milk used for cheese-making. The microbiota in the ripened cheese was generally dominated by starter LAB, while the RA of NSLAB was generally low. The dominant bacteria ASVs present in the ripened cheese were also found in farm bulk and dairy silo milk and in EPS, while some minor bacteria ASVs were only identified in EPSs and cheese, or only in cheese, suggesting that the dairy environment likely enriched such bacteria. Based on these results, the focus in future studies should be shifted from the milk microbiota to the effects of process variables and the dairy environment on cheese maturation.

## Figures and Tables

**Figure 1 foods-12-03796-f001:**
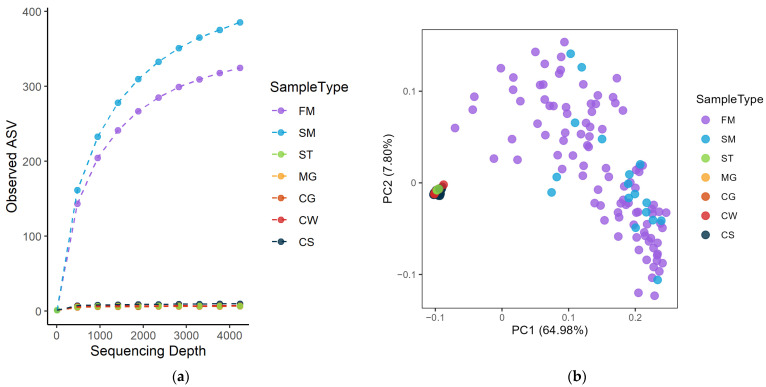
(**a**) Rarefaction curves of observed amplicon sequence variants (ASVs) and (**b**) principal coordinate analysis plot of generalised UniFrac distance associated with farm bulk milk (FM), dairy silo milk (SM), starter culture (ST), milk gel (MG), cheese grains (CG), fresh cheese aged for 24 h (CW) and cheese samples ripened for 7–20 months (CS).

**Figure 2 foods-12-03796-f002:**
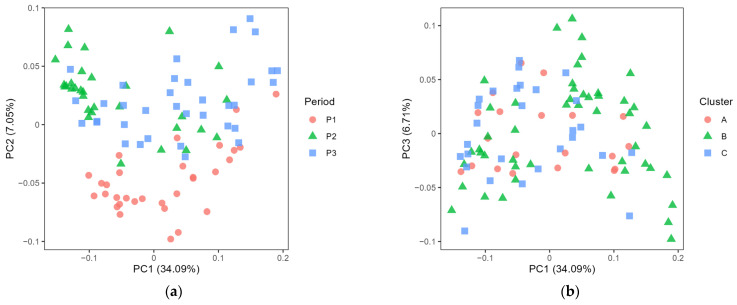
Principal coordinate analysis plot of generalised UniFrac distance comparing the microbiota of farm bulk milk according to (**a**) the different cheese production periods (P1–P3) and (**b**) the different farm clusters (A–C).

**Figure 3 foods-12-03796-f003:**
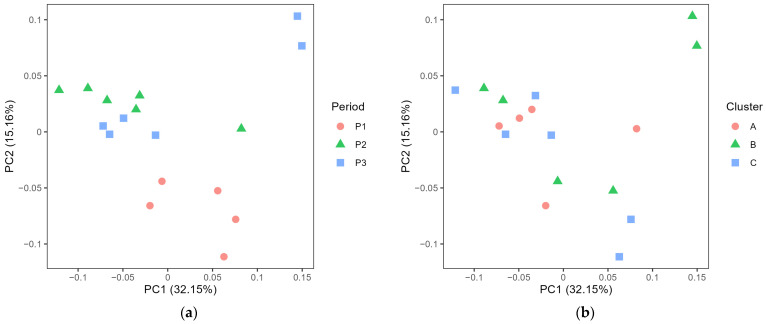
Principal coordinate analysis plot of generalised UniFrac distance comparing the microbiota of dairy silo milk according to (**a**) the different cheese production periods (P1–P3) and (**b**) the different farm clusters (A–C). One of the dairy silo milk samples for period 1, cluster A, is missing.

**Figure 4 foods-12-03796-f004:**
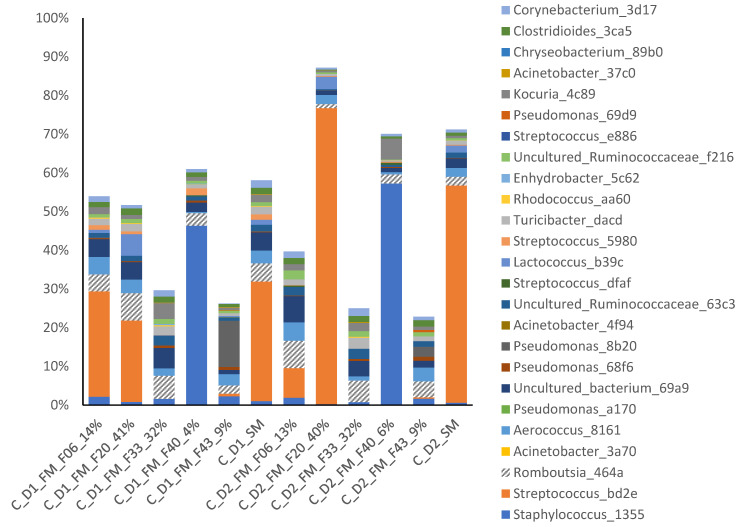
Relative abundance (RA, %) of the top 20 ASVs in bulk milk (FM) samples from the five farms (F06, F20, F33, F40 and F43) in cluster C, and the corresponding dairy silo milk (SM) samples on cheese-making days 1 and 2 (D1, D2) in period 1. The percentage given in the sample labels is the relative proportion of bulk milk from each individual farm out of the total milk volume in the dairy silo.

**Figure 5 foods-12-03796-f005:**
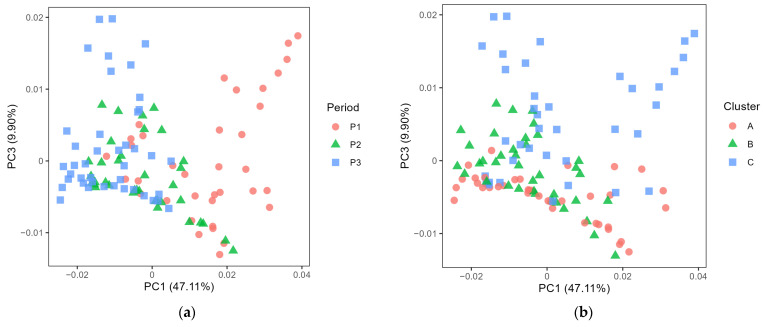
Principal coordinate analysis plot of generalised UniFrac distance comparing the microbiota in different cheeses depending on (**a**) cheese production period (P1–P3) and (**b**) farm cluster (A–C).

**Figure 6 foods-12-03796-f006:**
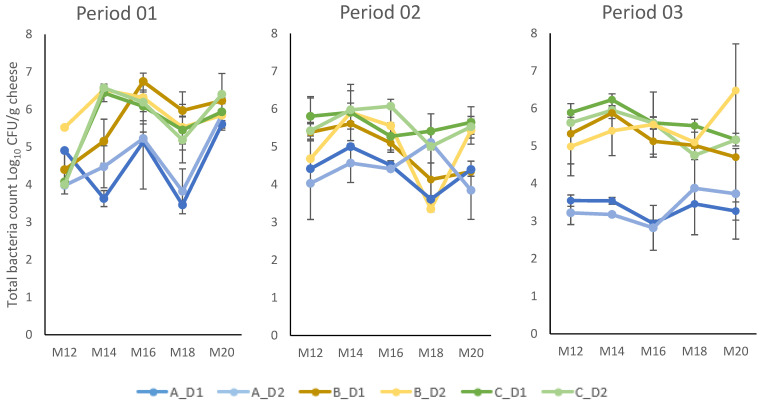
Total bacteria count in cheese aged for 12 to 20 months (M12–M20). The cheeses were produced using dairy silo milk from three different farm clusters (A–C) in a full-scale production setting. Cheese production was performed in three periods (P1–P3) over one year, and within each period milk from the three farm clusters was collected, and cheese production was performed, on two separate days (D1, D2).

**Figure 7 foods-12-03796-f007:**
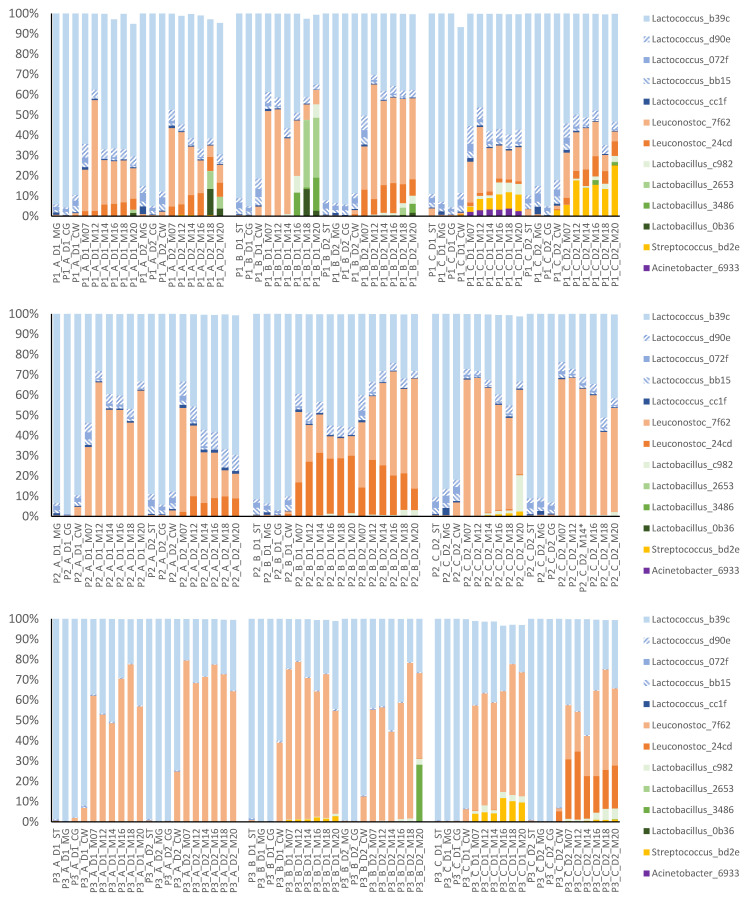
Relative abundance (RA, %) of the top 13 ASVs present in early process and cheese samples. Cheese production was performed in three periods (P1–P3) over one year. Within each period, milk from three farm clusters (A–C) was collected, and cheese was produced on two separate days within one week (D1, D2). Samples were taken at different time points in the cheese production process, including starter culture (ST), milk gel (MG), cheese grains (CG), fresh cheese at 24 h (CW) and cheese after ripening for 7 up to 20 months (M07–M20).

**Figure 8 foods-12-03796-f008:**
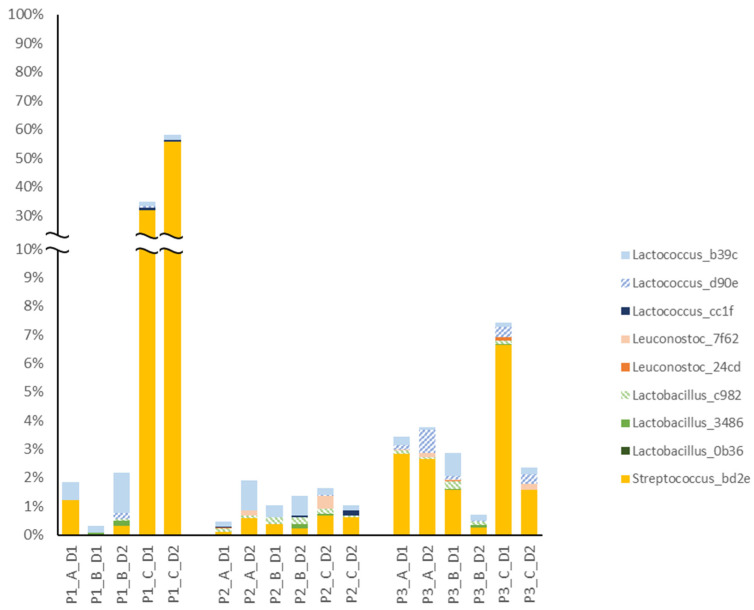
Relative abundance (RA, %) in dairy silo milk samples of nine ASVs that were among the top 13 ASVs found in cheese and also identified in the silo milk. Cheeses were produced using dairy silo milk from three different clusters of farms (A–C) in a full-scale production setting in three periods (P1–P3) distributed over one year. In each period, cheese was produced on two separate days (D1, D2). One of the silo milk samples for period 1, cluster A, day 2 (D2) is missing.

**Table 1 foods-12-03796-t001:** Milk selection criteria and basic characteristics associated with farms in clusters A–C. Farms belonging to cluster B had significantly smaller herd size than farms in clusters A and C and fed a significantly higher proportion of silage as round bales than farms in cluster A.

Farm Cluster and Selection Criteria	Number of Farms	Average Number of Cows per Herd (Min–Max) ^1^	Dominant Cow Breed ^2^ on the Farm (Number of Farms)	Milking System (Number of Farms)	Proportion of Silage Fed as Round Bales ^1^
A: Average milk quality	4	110 ^a^ (68–180)	SH (3) Mixed (1)	AMS (3) Milking parlour (1)	29% ^b^
B: Higher fat and protein content, fewer clostridia in milk	9	49 ^b^ (17–90)	SH (1) SRB (2) SKB (1) SJB (1) Mixed (4)	Tie-stall (6) AMS (2) Milking parlour (1)	73% ^a^
C: Higher total bacteria number, higher abundance of lactic acid bacteria in milk	5	84 ^a^ (41–127)	SH (4) SKB (1)	AMS (3) Milking parlour (2)	60% ^ab^

^1^ Values within columns with different superscript letters differ significantly (*p* < 0.05). ^2^ Dominant cow breed, i.e., >70% of cows in the herd: SH = Swedish Holstein, SRB = Swedish Red, SKB = Swedish Mountain breed or SJB = Swedish Jersey. AMS = automatic milking system.

**Table 2 foods-12-03796-t002:** Top 30 amplicon sequence variants (ASVs) and their relative abundance (RA) in 18 batches of cheese aged for 7 to 20 months, and number of batches in which ASV found in cheese were also detected in the corresponding farm bulk milk (FM), dairy silo milk (SM) and early process samples (EPS).

	Batches with ASV Found in Cheese and in Samples from Early Stages in the Process			Average RA (%) in Cheese Sampled after Ageing for 7–20 Months
ASV	FM	SM ^1^	EPS ^2^	
*Lactococcus* b39c	17	17	17	42.1
*Leuconostoc* 7f62	10	6	17	41.7
*Leuconostoc* 24cd	11	2	15	6.1
*Lactococcus* d90e	16	8	17	3.2
*Streptococcus* bd2e	18	16	9	1.8
*Lactobacillus* c982	17	12	11	1.3
*Lactobacillus* 2653	0	0	0	0.75
*Lactobacillus* 3486	6	4	2	0.61
*Lactococcus* 072f	0	0	14	0.52
*Lactococcus* bb15	0	0	11	0.51
*Lactococcus* cc1f	8	5	11	0.45
*Lactobacillus* 0b36	1	0	0	0.34
*Acinetobacter* 6933	0	0	1	0.17
*Streptococcus* 0c46	2	1	1	0.15
*Lactobacillus* bf19	0	0	1	0.13
*Lactobacillus* 57c1	1	0	0	0.064
*Lactococcus* 6fa5	0	0	0	0.041
*Romboutsia* 464a	15	14	0	0.013
*Tetragenococcus* 49a8	1	0	0	0.011
*Turicibacter* dacd	12	11	0	0.011
*Paeniclostridium* 69a9	14	13	0	0.010
*Streptococcus* 84e3	1	1	0	0.010
*Lactobacillus* 61ee	2	2	1	0.008
*Staphylococcus* 1355	10	9	0	0.005
*Clostridium* ddf8	0	0	1	0.005
*Trueperella* b116	4	3	0	0.005
*Enterococcus* 028e	5	3	0	0.004
*Lactococcus* ffd9	0	0	0	0.003
Unclassified Enterobacteriaceae 0315	2	1	2	0.002
Unclassified Enterobacteriaceae 0942	0	0	1	0.0004

^1^ One silo milk sample missing. ^2^ EPS missing for one batch. EPS included freshly propagated starter culture, milk gel and cheese grains before pressing.

## Data Availability

Data are contained within the article.

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
