# Peer review of "Associations between the Bacterial Composition of Farm Bulk Milk and the Microbiota in the Resulting Swedish Long-Ripened Cheese"

_foods, 2023, doi:10.3390/foods12203796_

Round 1

Reviewer 1 Report

It was a long-term, labor-intensive work. The purpose of the research and the desired conclusion are clear. The methods used are up to date but I recommend making various minor changes:

Title. Please include bulk milk in the title and use ‘bacterial composition’ instead of microbial composition. and I recommend using the expression ‘Swedish cheese’ in the title.

Line 60. These bacteria should be written once in clear form when they are first written. For example, Lacticaseibacillus casei or Lactiplantibacillus plantarum. Also, please write the abbreviation Lb with a period next to it.

Line 78. It would be better not to use the words composition and community at the same time.

Line 273. log CFU/g ??

Line 359. As ripening begins and continues, species that cannot compete are eliminated and alpha diversity decreases. The process works like this in the fermentation microbiota. As the food/product waits, the species that succumb to competition disappear from the environment. As is the case with the alpha diversity between farm milk and silo milk.You can also write about this situation regarding the lower alpha diversity values of ripened cheese compared to fresh cheese.

Line 399. This information about the presence of acinetobacter in camembert cheese is irrelevant here and disrupts the flow of narrative.

Line 418. In ripened cheeses, the number and relative abundance of lactococcus species are generally affected by acidity, etc. They cannot withstand the conditions any longer and decreases. Please add an explanation about this here.

Line 451. Starter LAB counts??

Lİne 454. This situation is also a disadvantage of this study. It is pointless to discuss whether any bacterial species transferred from bulk milk or farm milk to cheese, since PMA was not applied.

Please add a comment or discussion on why the genera staphylococci and rhomboutsia, which have high relative abundance in raw milk, are not among the most dominant genera in cheese.

Reviewer 2 Report

General comments:

A very interesting study! Aiming to unveil a specific concern to be considered by all cheese producers.

Specific comments:

Lactobacillus taxonomy has been comprehensively revised, please alter throughout the document, including data analysis.

Regarding figure 4: In my opinion it would be best if instead of ASVs the authors decided if they prefer to include genera information, as an alternative to a genus and species mix. Example: Combine data from Streptococcus_dysgalactiae_5980 and Streptococcus_dfaf (the same for Pseudomonas…). Yes, I am aware that they may distinct microorganisms, but I truly believe that mixing genera and species levels in the same data display does not work. You should choose the preferred taxonomical level, which undoubtedly has to be genus because you do not have much information on species allocation.

I understand why you need ASVs for proper results discussion, the previous idea concerns only figure 4.

Was microbial isolation performed in parallel, for the samples used for DNA extraction? This approach would undoubtedly provide relevant information of the specificities of the in house microbiota. If not considered, this “limitation” should be commented on the document.

Moreover, were the cheeses evaluated by a sensory panel? And what about sampling cheesemaking settings to further allocate the in house microbiota to specific locations?

Overall, the manuscript needs further discussion on the study’s limitations and specifics on future perspectives. How can this information be relevant to the cheese producers? How can they optimize the cheese making processes, in order to produce more standardized products and attain a better control of the ripening period?

Reviewer 3 Report

The publication deals with an interesting topic about indigenous types of dairy products. I have the following notes:
1. How was the location of the farms from which the milk was taken determined?
2. If possible, please present a technological scheme for the production of the relevant cheeses.
3. What is the influence of climatic factors on the microbiota?
